

# Food safety in Thailand 4: comparison of pesticide residues found in three commonly consumed vegetables purchased from local markets and supermarkets in Thailand

Sompon Wanwimolruk[1], Kamonrat Phopin[1,2], Somchai Boonpangrak[1] and Virapong Prachayasittikul[2]

[1] Center for Research and Innovation, Faculty of Medical Technology, Mahidol University, Bangkok, Thailand
[2] Department of Clinical Microbiology and Applied Technology, Faculty of Medical Technology, Mahidol University, Bangkok, Thailand

Corresponding author
Sompon Wanwimolruk,
sompon-999@hotmail.com,
sompon.wan@mahidol.ac.th

## ABSTRACT

**Background**. The wide use of pesticides raises concerns on the health risks associated with pesticide exposure. For developing countries, like Thailand, pesticide monitoring program (in vegetables and fruits) and also the maximum residue limits (MRL) regulation have not been entirely implemented. The MRL is a product limit, not a safety limit. The MRL is the maximum concentration of a pesticide residue (expressed as mg/kg) recommended by the Codex Alimentarius Commission to be legally permitted in or on food commodities and animal feeds (*Codex Alimentarius Commission, 2015*; *European Commission, 2015*). MRLs are based on supervised residue trial data where the pesticide has been applied in accordance with GAP (Good Agricultural Practice). This study aims at providing comparison data on pesticide residues found in three commonly consumed vegetables (Chinese kale, pakchoi and morning glory) purchased from some local markets and supermarkets in Thailand.

**Methods**. These vegetables were randomly bought from local markets and supermarkets. Then they were analyzed for the content of 28 pesticides by using GC-MS/MS.

**Results**. Types of pesticides detected in the samples either from local markets or supermarkets were similar. The incidence of detected pesticides was 100% (local markets) and 99% (supermarkets) for the Chinese kale; 98% (local markets) and 100% (supermarkets) for the pakchoi; and 99% (local markets) and 97% (supermarkets) for the morning glory samples. The pesticides were detected exceeding their MRL at a rate of 48% (local markets) and 35% (supermarkets) for the Chinese kale; 71% (local markets) and 55% (supermarkets) for the pakchoi, and 42% (local markets) and 49% (supermarkets) for the morning glory.

**Discussion**. These rates are much higher than those seen in developed countries. It should be noted that these findings were assessed on basis of using criteria (such as MRL) obtained from developed countries. Our findings were also confined to these vegetables sold in a few central provinces of Thailand and did not reflect for the whole country as sample sizes were small. Risk assessment due to consuming these pesticide contaminated vegetables, still remains to be evaluated. However, remarkably high incidence rates of detected pesticides give warning to the Thai authorities to implement

proper regulations on pesticide monitoring program. Similar incidence of pesticide contamination found in the vegetables bought from local markets and supermarkets raises question regarding the quality of organic vegetables domestically sold in Thailand. This conclusion excludes Thai export quality vegetables and fruits routinely monitored for pesticide contamination before exporting.

# INTRODUCTION

An enormous concern on toxic pesticides in foods has been raised because of its negative health and environmental impacts. This is due to the widespread use of pesticides in agriculture. The main exposure to pesticides for humans is oral ingestion, especially by vegetables and fruits (*Claeys et al., 2008*; *Drouillet-Pinard et al., 2011*). Toxicity and human health risk associated with pesticide contamination in foods has made it necessary to regulate pesticide residues in our foods (*Cervera et al., 2014*). Detection and quantification of pesticide residues in food samples are essential to verify whether these pesticides are within limits, so called ''maximum residue limits (MRL)''. This regulation was established by the European Commission and other regulatory authorities. Many developed countries have approved this regulation to oversee and operate their food safety affairs. In contrast in developing countries such as Thailand, good agricultural practices (GAP) have not fully been implemented, nor has a successful pesticide monitoring program. The exception is for produce that will be exported. Pesticides have been greatly used in agriculture in Thailand (*Harnpicharnchai, Chaiear & Charerntanyarak, 2013*). The most popular classes of pesticides imported into Thailand are herbicides, followed by insecticides and fungicides (*Sapbamrer & Nata, 2014*). Among the insecticides, organophosphates and carbamates are very commonly used for protecting crops from insect invasion. The use of pesticides in agriculture has been linked with occupational health of farmers, gardeners and consumers (*Chan, 1990*; *Sapbamrer & Nata, 2014*).

Chinese kale (*Brassica oleracea*) is also known as Chinese broccoli. Chinese kale is a leaf vegetable appearing thick and flat, with glossy blue–green leaves, thick stems and a small number of tiny, almost vestigial flower heads similar to those of broccoli. The flavour of Chinese kale is very like to that of broccoli, but somewhat more bitter. Chinese kale is used extensively in Chinese cuisine, and especially in Cantonese cuisine. In Thailand, a number of admired Thai dishes have Chinese kale as a principal ingredient. In some dishes, Chinese kale is consumed fresh, without cooking. This possesses potential for toxicity if the vegetables are eaten fresh every day without washing them properly. Pakchoi [*Brassica chinensis Jusl var parachinensis* (Bailey) Tsen & Lee] is a species in the Brassicaceae which is a popular vegetable consumed in Thailand, as well as in Southeast Asia and southern China. Unlike napa cabbage (*Brassica pekinensis*), pakchoi does not form heads; instead, it has smooth, dark green leaf blades forming a cluster reminiscent of mustard or celery.

Water morning glory (*Ipomoea aquatic* Forsk) is a semiaquatic, tropical plant grown as a vegetable in East, South and Southeast Asia. It is also known as water spinach, water convolvulus, or by the more ambiguous names Chinese spinach, Chinese convolvulus or swamp cabbage (*Nagendra Prasad, Shivamurthy & Aradhya, 2008*). It is known as pak bung in Thai, ong choy in Chinese and kangkong in Tagalog. Water morning glory is one of the most popular vegetables in Thai, Burmese, Lao, Cambodian, Malay, Vietnamese, Filipino, and Chinese cuisines.

Pesticide residues have been found in many raw agricultural commodities such as vegetables and fruits, and processed foods worldwide in the past decades (*Chen et al., 2011*; *Chen et al., 2014*; *Huan et al., 2015*; *Li et al., 2014*; *Osei-Fosu et al., 2014*; *Sapbamrer & Hongsibsong, 2014*; *Wang et al., 2013*; *Wanwimolruk et al., 2015a*; *Wanwimolruk et al., 2015b*). Presently, information on pesticide contamination in vegetables in Thailand is limited and systemic investigation is desired to verify the current status of pesticide contamination in food, particularly in vegetables and fruits. Also, organic fruits and vegetables currently sold in Thailand displays to consumers with no confidence in regard to quality whether the produce is pesticide-free. Many supermarkets have placed labels on fruits and vegetables implying that they are either organically grown or pesticide-free. The supermarkets in Thailand sell not only organic produce. The fresh vegetables sold in supermarkets in Thailand can be categorized into four groups, i.e., conventional, organic, pesticide-free and safe vegetables. Regarding the latter category, the Thai people have questioned if they are pesticide-free or organic vegetables. Consequently, people are prepared to buy vegetables and fruits from supermarkets at much higher price than those from local markets. This is because they have a high expectation that supermarket produce is safe from pesticide contamination. However, there is no scientific-based evidence to verify the supermarkets' claims and people's beliefs. Therefore, the purpose of this study was to provide comparison data on pesticide residues found in three commonly consumed vegetables (Chinese kale, pakchoi and water morning glory) purchased from local markets and supermarkets. This study was not intended to compare the conventionally grown and organically grown vegetables. However, the present study raises questions about the quality of organically grown vegetables in Thailand.

## MATERIALS AND METHODS

### Chemicals and standards

Anhydrous magnesium sulphate, sodium chloride, primary and secondary amine (PSA, particle size 40 μm), graphite carbon black (GCB) and C18 sorbent (particle size 40 μm) were obtained from Supelco (Sigma-Aldrich Corp., St. Louis, USA). HPLC-grade acetonitrile was purchased from Merck (Darmstadt, Germany). Twenty eight pesticides and two metablolite standards including aldrin, atrazine, captan, carbaryl, carbofuran (and its two metabolites carbofuran-3-hydroxy and carbofuran-3-keto), carbosulfan, chlormefos, chlorpyrifos, chlorothalonil, λ-cyhalothrin, cypermethrin, deltamethrin, diazinon, dichlorvos, dicofol, dimethoate, ethion, fenitrothion, fenvalerate, malathion, metalaxyl, methidathion, methomyl, paraoxon-methyl, phosalone, pirimicarb, pirimiphos-methyl and profenofos were purchased from Dr. Ehrenstorfer (Augsburg, Germany). Purity

of these pesticide standards was >98%. Individual stock of standard solutions (1,000 mg/L) was prepared in acetonitrile.

## Vegetable samples

Three vegetables were selected for this study: Chinese kale, pakchoi and water morning glory. The selection was based on their high consumption in Thailand. These three vegetables are widely consumed among Thai and Asian people. Chinese kale samples ($n = 137$) were purchased randomly from local open-air markets ($n = 69$) and supermarkets ($n = 68$). For pakchoi, a total of 125 samples were bought from local markets ($n = 63$) and supermarkets ($n = 62$). Samples of water morning glory ($n = 135$) were purchased randomly from local markets ($n = 74$) and supermarkets ($n = 61$). These markets were located in central provinces of Thailand including Bangkok, Nakhon Pathom, Nonthaburi, Ayutthaya, Pathumthani, Samutsakorn and Nakhon Ratchasima. These provinces are located surrounding Bangkok, Thailand, within a radial distance of 260 km. The supermarkets which the vegetable samples were bought from were Big C, Foodland, Jiffy Plus, Lemon Farm, Max Valu, Tesco Lotus, Tops and Villa Market. The study was carried out over a year from November 2013 to December 2014. At the local markets from which vegetable samples were bought, the produce that was for sale came from conventional farms and was not claimed to be 'organic produce'. The reason we purchased conventional from the local markets was because almost all produce sold in the local markets are known to be conventionally grown. There are only a few produce sold in the local markets that are labelled as organic, whereas the vegetable samples purchased from supermarkets were sorted into four groups, i.e., conventional, organic, pesticide-free and safe vegetables. Most of vegetable samples bought from the supermarkets were claimed to be organic and pesticide-free vegetables. Like for our previous study (*Wanwimolruk et al., 2015b*), it is very difficult to get the information on the suppliers as most of the workers in both supermarkets and local markets had no idea where the vegetables were bought from. Therefore, we could not obtain accurate information on suppliers. Approximately 500 g of vegetables were purchased and the samples were transported to the laboratory for analysis which was done within 24 h. The representative portion (150–200 g) of the vegetable sample was chopped into tiny pieces and homogenized using a food processor and mixed carefully. The homogenized samples were then extracted and treated as described in the following section.

## Sample preparation

The analysis of pesticide residues was performed using the pesticide multiresidue QuEChERS (Quick Easy Cheap Effective Rugged and Safe) method as explained previously (*Anastassiades et al., 2003*; *Lehotay, 2007*; *Lehotay et al., 2010*; *Paya et al., 2007*). Briefly, extraction of pesticides was performed by extracting 15 g of homogenized vegetable with 15 ml acetonitrile saturated with 6 g of magnesium sulphate and 1.5 g of sodium chloride. This extraction process was pursued by a cleaning up procedure. This was achieved by transferring the supernatant (1 mL) into another tube comprising 50 mg of primary-secondary amine (PSA), 7.5 mg graphite carbon black (GCB) and 150 mg magnesium sulphate. After shaking and centrifugation, the extract supernatant was then transferred to an autosampler vial for direct injection into the Bruker GC/MS/MS system.

## GC-MS/MS analysis

Detection of pesticides was accomplished by using a Bruker 456 gas chromatography (GC) coupled with Bruker Scion Triple Quadrupole mass spectrometer (GC-MS/MS). Details of GC-MS/MS conditions were referred to as in the previous reports (*Duff & Voglino, 2012*; *Wanwimolruk et al., 2015b*). Multiple reaction monitoring (MRM) acquisition method and two ion transition at the experimentally optimized collision energy (CE) were monitored for each pesticide analyte.

## Calibration and quantification

A working surrogate spiking standard solutions of pesticides were made by an appropriate dilution of the stock solutions with acetonitrile. These standard solutions were guarded from light and kept frozen at $-20$ °C until required. Calibration curves of each pesticide of interest were conducted using an internal standard method according to the established procedure (*Koesukwiwat, Lehotay & Leepipatpiboon, 2011*; *Lehotay, 2007*; *Lehotay et al., 2010*; *Wanwimolruk et al., 2015b*). These were conducted using the same procedure each time when a new unknown sample set was analyzed. Aldrin was used as an internal standard. The ratio of the peak area of the pesticide standard to that of the internal standard was employed for quantification. Recovery studies for method validation were conducted as previously described (*Koesukwiwat, Lehotay & Leepipatpiboon, 2011*; *Wanwimolruk et al., 2015b*). The method validation in regard to reproducibility, calibration linear range, limit of detection (LOD), limit of quantification (LOQ) was performed for each vegetable matrix as expressed previously (*Dong et al., 2012*; *Koesukwiwat, Lehotay & Leepipatpiboon, 2011*). Quantitation of pesticides in an unknown vegetable sample was carried out in duplicate unless otherwise stated. MRL values for each pesticide in the vegetable of interest were quoted from recommended MRL values established by *Thailand Ministry of Agriculture Cooperation (2013)*, *Codex Alimentarius Commission (2015)* and *European Commission (2015)*. These three references were used because not all pesticides' MRLs were listed in the individual reference.

## Data treatment

Vegetable samples were grouped into two categories, according to the sources where the samples were purchased, i.e., local markets and supermarkets. Pesticide concentrations obtained from the GC-MS/MS analysis were treated separately for each vegetable studied. These data were further evaluated to determine % total detection of pesticide residues, % of samples which pesticides were not detected, % of samples contained pesticide residues of <MRL, and % of samples contained pesticide residues of >MRL. For each vegetable of interest the number of samples (or frequency) containing individual pesticide was counted with aid of using Excel Microsoft program. Also the bar graphs were plotted (Excel) from these data to show frequency distribution or bar graphs illustrating types of pesticides in Chinese kale, pakchoi and morning glory, separately. Numbers of samples containing pesticide residues of >MRL were determined by using the MRL reference values for each pesticide and for particular commodity. These were also performed by using Excel.

## Statistical analysis

All results are presented as either mean $\pm$ standard deviation (S.D.) or median. The differences of parameter between two sample groups were assessed by either unpaired Student's $t$-test or the Mann–Whitney $U$-test, depending on their normality of distribution. The statistical significance level was customary to $P < 0.05$. All statistical analyses were assessed using the software SPSS statistical package for Windows version 18.0 (SPSS Inc., Chicago, IL, USA).

## RESULTS

The GC-MS/MS method was validated to determine efficiency and accuracy of the analytical assay. Excellent linearity of calibration curves of each pesticide standards were attained as illustrated by the coefficient of determination ($r^2$) values of >0.92. For instance, for Chinese kale, the linearity of calibration curves of all twelve pesticides detected (e.g., cypermethrin, deltamethrin, diazinon, dimethoate, metalaxyl, and profenofos) were excellent with $r^2 > 0.93$. When the pesticides of interest were assayed at 0.01 ppb the signal-to-noise ratio was well above 30 for all pesticides studied. Therefore, detection limits were below 0.01 ppb using the sample preparation procedures described previously. The precision of the method was verified by the reproducibility of the retention time and peak area. It was noticed that the retention time and peak area of all pesticides were in good precision. The relative standard deviations (RSD) of repeatability for cypermethrin and metalaxyl in Chinese kale samples were 5.2% and 4.6%, respectively. While the RSD of reproducibility for cypermethrin and metalaxyl in Chinese kale samples were 12.3% and 9.1%. Overall, their relative RSD of repeatability were lower than 8% whereas the RSD of reproducibility were lower than 17%. In general, the mean recoveries of all pesticides studied from fortified samples in five replicated experiments were in the range of 75–114%. For example, the mean recovery of carbaryl and metalaxyl in Chinese kale were $102 \pm 11\%$ and $97 \pm 7\%$ at a concentration of 100 ppb. These ranges of recovery fall within the typical acceptance criteria for quantitative regulatory methods (*Koesukwiwat, Lehotay & Leepipatpiboon, 2011*). Similar observations of assay validations were found with respect to other two vegetables studied, i.e., pakchoi and morning glory.

Twenty eight pesticides studied were selected on the basis of their widespread use in agriculture in Thailand. Although the most popular classes of pesticides imported into Thailand are herbicides (*Sapbamrer & Nata, 2014*), only one herbicide namely atrazine was studied. This was simply due to lack of budget to purchase other pesticide standards. Glyphosate is very commonly used herbicide in Thailand but the analytical assay for glyphosate is rather expensive and not yet available in our laboratory. The GC-MS/MS method employed in this study offered satisfactorily separation with high sensitivity and selectivity for quantitation of all 28 pesticides of interest (*Wanwimolruk et al., 2015b*). The absence of co-extracted interferences for all varieties of leaf vegetables, Chinese kales, pakchoi and water morning glory, was demonstrated by blank extract analysis showing there was no interfering peak co-eluted with analytes of interest. Moreover, in all vegetable samples tested, there were no identifiable peaks detected with the same retention time as aldrin

(retention time = 16.02 min) that was used as an internal standard in our GC-MS/MS assay. This supports the rationality of employing aldrin as the internal standard for the assays.

Of 28 pesticides tested, 12 pesticides were detected in the Chinese kale samples purchased from supermarkets (Fig. 1). These included carbaryl, carbofuran, chlorothalonil, chlorpyrifos, λ-cyhalothrin, cypermethrin, deltamethrin, diazinon, dimethoate, malathion, metalaxyl and profenofos. Nevertheless, chlorothalonil and deltamethrin were not detected in those Chinese kale samples purchased from local markets (Fig. 1), while malathion was not found in the samples bought from supermarkets. Most of Chinese kale samples (88% in local markets, 91% in supermarket samples) had multiple pesticide residues. Overall, metalaxyl, dimethoate and diazinon appeared to be the most often found pesticides in the Chinese kale samples from both sources (Fig. 1). The occurrence rate of metalaxyl in the local market samples was 91% (63/69) and was 94% (64/68) for the supermarket samples. However, none of the Chinese kale samples purchased from both local markets and supermarkets had metalaxyl that exceeded the recommended MRL value (2,000 ppb). Rates of occurrence for dimethoate in the Chinese kale samples were 80% (55/69) and 88% (60/68) for the Chinese kale samples from local markets and supermarkets, respectively. Of 69 samples from local markets, 23 of them had dimethoate exceeding the MRL value (20 ppb). This corresponds to a rate greater than dimethoate's MRL of 33%. Eleven samples purchased from the supermarkets were found to contain dimethoate that exceeded the MRL. These samples exceeded dimethoate's MRL by 16%.

Diazinon was other commonly pesticide detected in the Chinese kale samples studied. It was detected in 62% (43/69) and 74% (50/68) of the Chinese kale samples from local markets and supermarkets, respectively (Fig. 1). None of the local market or supermarket samples had diazinon levels that exceeded the recommended MRL (50 ppb). Three other pesticides were also detected in the Chinese kale samples which were profenofos, cypermethrin and carbaryl. Profenofos was detected in the Chinese kale samples from both sources, with moderate occurrence rates of 33% (23/69) for the local market samples and 29% (20/68) for the supermarket samples. Eleven of the samples purchased from the local markets had profenofos levels exceeding the MRL (10 ppb), whereas twelve samples from the supermarkets contained profenofos at concentrations greater than the recommended MRL. Of note, profenofos concentrations detected in the Chinese kale samples was found to vary widely among the samples from both sources with a range from 0.1–2,095 ppb. Pyrethroid pesticide cypermethrin was also detected in the Chinese kale samples at a relatively low rate of detection. Cypermethrin was found in 16% (11/69) of the samples bought from the local market, similarly 15% (10/68) of the supermarket samples contained cypermethrin (Fig. 1). One of the local market samples had cypermethrin that exceeded the MRL, while all of the supermarket samples (3 samples) were found to have cypermethrin that exceeded the MRL value (1,000 ppb). Carbaryl was detected in 19% (13/69) and 15% (10/68) of the Chinese kale samples from local markets and supermarkets, respectively (Fig. 1). Levels of carbaryl found in these samples ranged from 0.1 to 606 ppb, in which two of the supermarket samples contained carbaryl exceeding its MRL value (50 ppb). The rest of pesticides found in the Chinese kale samples were detected with relative low

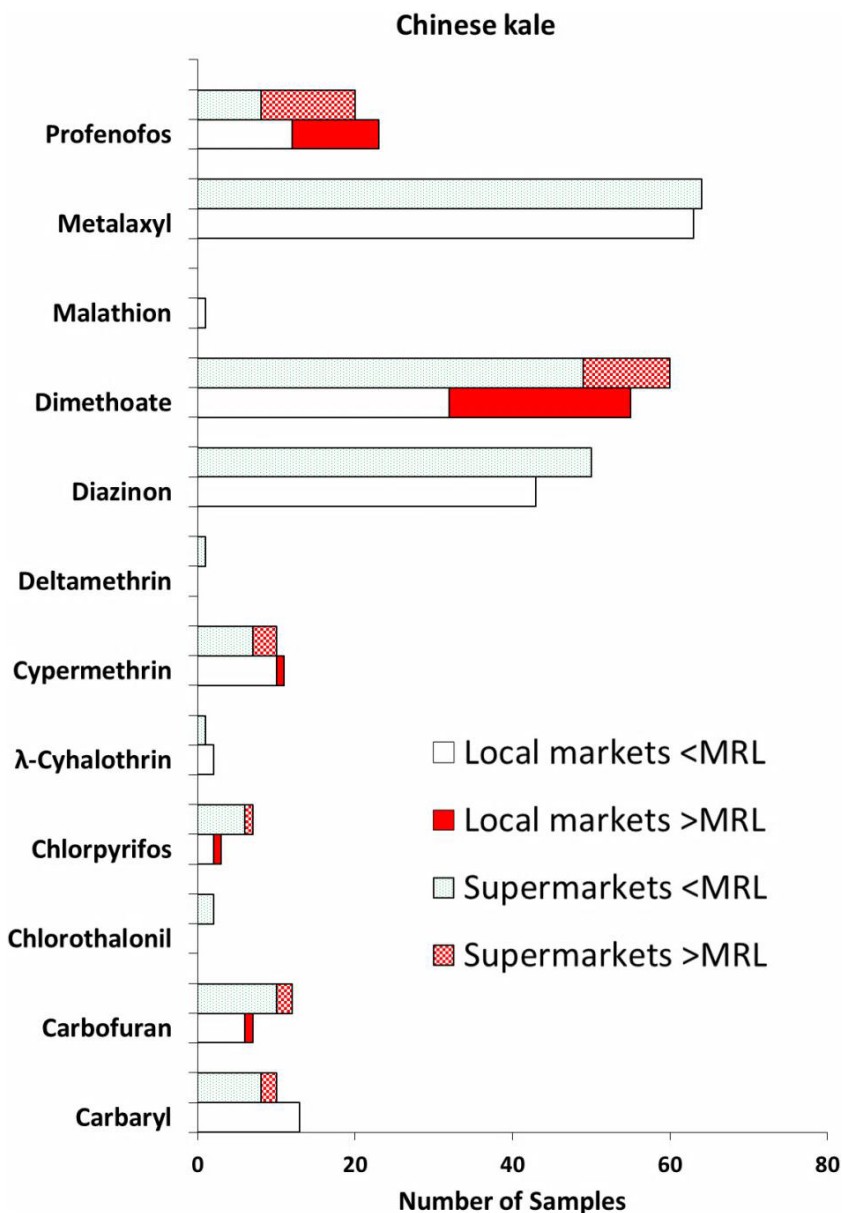

**Figure 1  Type of pesticides detected in the Chinese kale samples purchased from the local markets** (*n* = 69) **and the supermarkets** (*n* = 68)**.** For each pesticide detected, the lower bars are for samples from the local markets, and the upper bars are for samples from the supermarkets.

rate of occurrence. These include carbofuran, chlorothalonil, chlorpyrifos, deltamethrin, λ-cyhalothrin and malathion.

The incidence of pesticide detection, i.e., the % of total pesticide detection in the Chinese kale samples from both sources were extremely high, that is 100% and 99% for the samples bought from the local markets and the supermarkets, respectively. Of interest, the incidence of pesticides detected exceeding the recommended MRL values was 48% in the Chinese kale samples purchased from the local markets. This was slightly higher than the
incidence of pesticide detected exceeding the MRL of 35% observed in the samples from the supermarkets. Very small samples were found to contain no pesticides; this represents a rate of free of pesticides of 1% in the supermarket samples.

Nine pesticides were detected in both the pakchoi samples purchased from the local markets and the supermarkets (Fig. 2). These were carbaryl, carbofuran, chlorpyrifos, λ-cyhalothrin, cypermethrin, diazinon, dimethoate, metalaxyl and profenofos. Similar to findings observed in the Chinese kale, three pesticides, metalaxyl, dimethoate and diazinon, were the most often detected in the pakchoi samples collected from both sources. Few pakchoi samples had only one pesticide whereas others (92% in local markets, 97% in supermarket samples) had multiple pesticide residues. Profiles of pesticide types detected in the pakchoi samples from both sources were similar. Like the Chinese kale, occurrence of metalaxyl in pakchoi samples was very high at 97% (61/63) for the samples purchased from the local markets, and 98% (61/62) of the samples from the supermarkets were found to have metalaxyl residues (Fig. 2). Among these local market samples, 13 samples (21%) had metalaxyl levels that exceeded the recommended MRL (50 ppb). For the samples bought from the supermarkets, 11 samples (18%) had metalaxyl that exceeded the MRL. Dimethoate was found in 94% (59/63) and 87% (54/62) of the pakchoi samples from local markets and supermarkets, respectively (Fig. 2). Thirty-four samples from the local markets (54%) had dimethoate levels of greater than the recommended MRL (20 ppb), whereas 23 supermarket samples (37%) had dimethoate that exceeded its recommended MRL. Rates of occurrence for diazinon in the pakchoi samples were 57% (36/63) and 65% (40/62) for the samples from local markets and supermarkets, respectively (Fig. 2). None of the pakchoi samples bought from both the local markets and the supermarkets had diazinon levels above the MRL (50 ppb). Carbofuran, chlorpyrifos and cypermethrin were detected in pakchoi samples from both the local market and supermarkets but with moderate occurrence rates. Cypermethrin was found in 19% (12/63) of the pakchoi samples bought from the local market samples, while 21% (13/62) of the supermarket samples contained cypermethrin (Fig. 2). Two of the pakchoi samples bought from the local markets were found to have cypermethrin exceeding the recommended MRL (1000 ppb). Five of the pakchoi samples bought from supermarkets had cypermethrin exceeding the MRL. Chlorpyrifos was detected in 11% (7/63) of the pakchoi samples purchased from the local markets, whereas 16% (10/62) of the supermarket samples were found to contain chlorpyrifos residues (Fig. 2). Two of the pakchoi samples purchased from the local markets and one supermarket sample had chlorpyrifos that exceeded the recommended MRL (1000 ppb). For carbofuran, the pesticide detection rate was 32% (20/63) in the local market samples, and 29% (18/62) in the supermarket samples. Even though other three pesticides including carbaryl, λ-cyhalothrin and profenofos were also detected in the pakchoi samples but the occurrence rates were relatively low (Fig. 2).

The total incidence of pesticide detection in the pakchoi samples was 98% and 100% for the samples bought from the local markets and from the supermarkets, respectively. The incidence of pesticides detected exceeding the recommended MRL values was 71% in the pakchoi samples purchased from the local markets. While the incidence of MRL exceedance was 55% in the pakchoi samples bought from the supermarkets. These left the

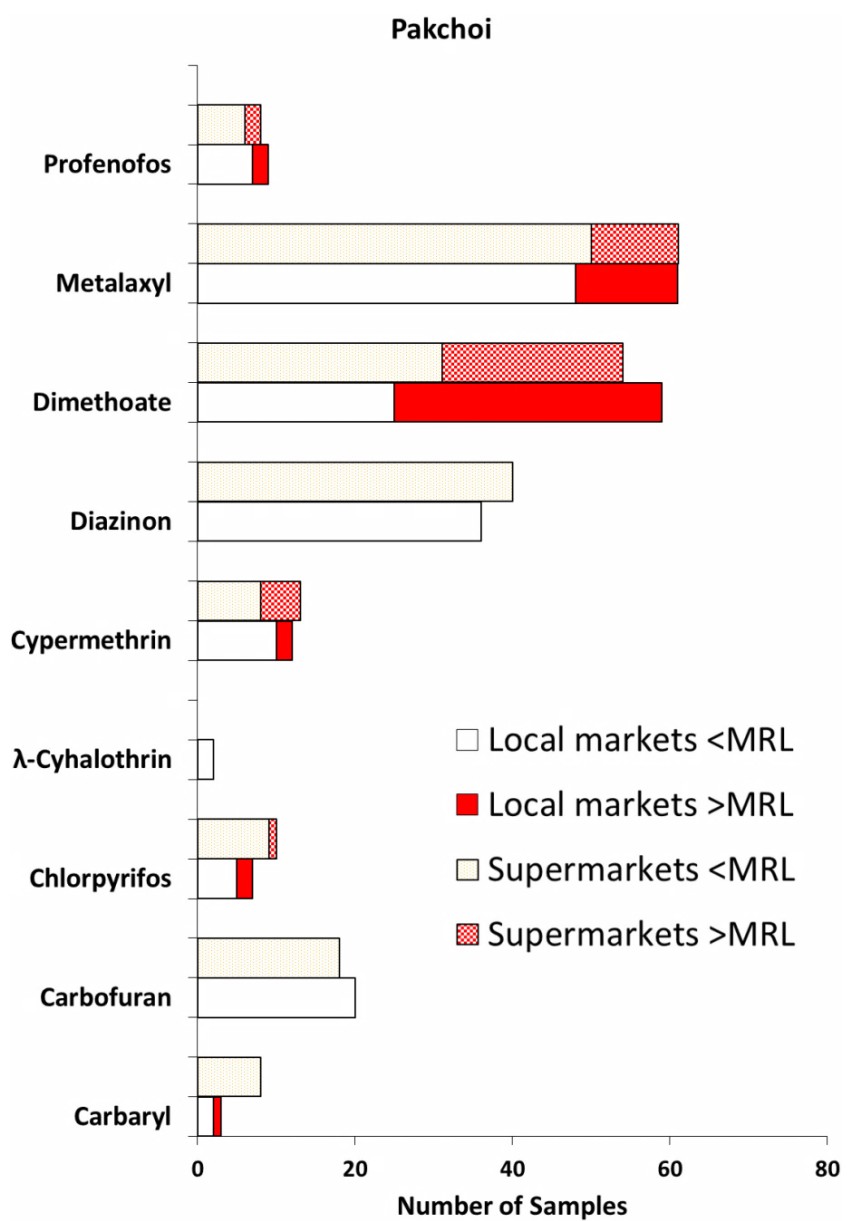

**Figure 2** **Type of pesticides detected in the pakchoi samples bought from the local markets ($n = 63$) and the supermarkets ($n = 62$).** For each pesticide detected, the lower bars are for samples from the local markets, and the upper bars are for samples from the supermarkets.

proportions of pakchoi samples having pesticide residues of less than MRL and without pesticides to be approximately 30%.

Of 28 pesticides investigated, 12 different individual pesticides were detected in the water morning glory samples purchased from both the local markets and the supermarkets (Fig. 3). Eight common pesticides detected in both the morning glory samples from the local markets and the supermarkets were carbofuran, chlorpyrifos, λ-cyhalothrin, cypermethrin, diazinon, dimethoate, metalaxyl and profenofos. Few samples contained

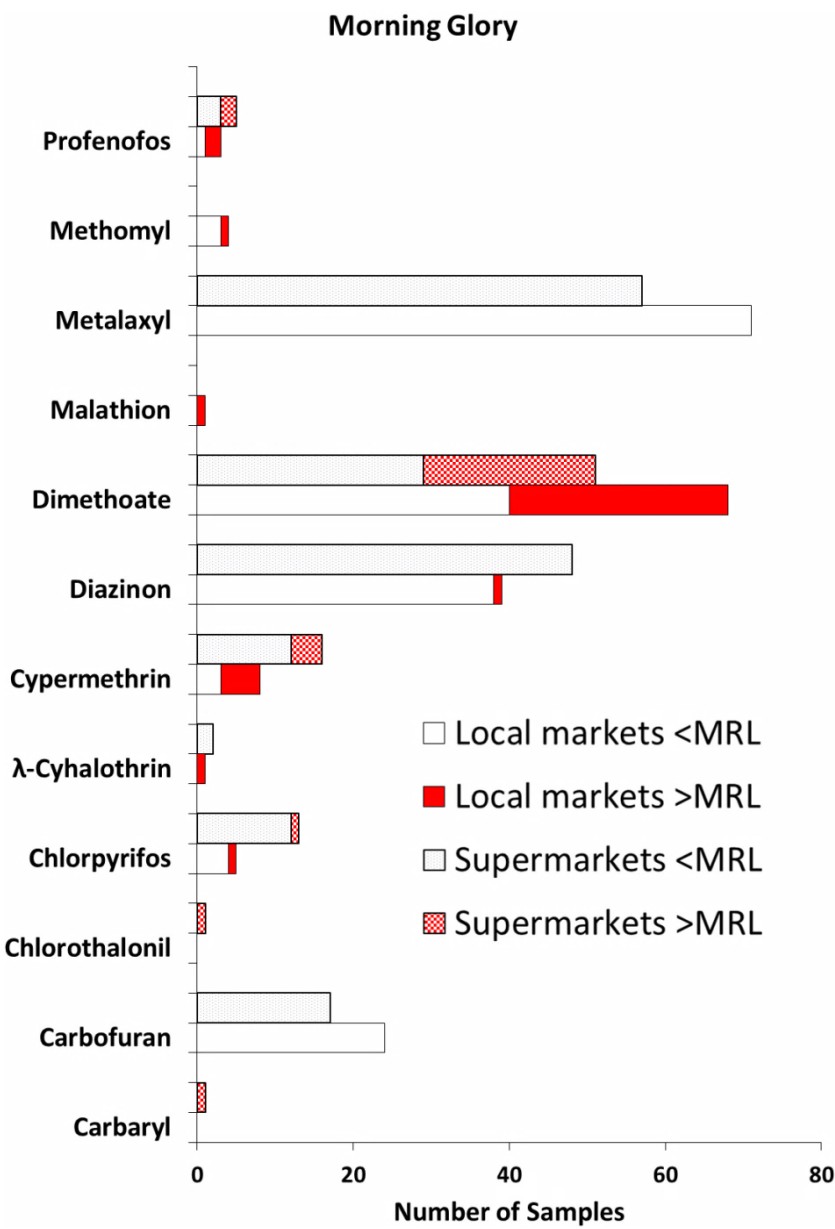

**Figure 3** **Type of pesticides detected in the morning glory samples bought from the local markets (*n* = 74) and the supermarkets (*n* = 61).** For each pesticide detected, the lower bars are for samples from the local markets, and the upper bars are for samples from the supermarkets.

only one pesticide, but most of them (90% in local markets, 89% in supermarket samples) had multiple pesticide residues. Again, similar to Chinese kale and pakchoi, metalaxyl, dimethoate and diazinon appeared to be the most often found pesticides in the water morning glory samples from both sources. Occurrence rates of metalaxyl in morning glory samples were 96% (71/74) and 93% (57/61) for the local market and the supermarket samples, respectively. All of the morning glory samples tested had metalaxyl levels below the recommended MRL (2,000 ppb). Occurrence rates for dimethoate in the water morning

glory samples were 92% (68/74) for the local market samples, and 84% (51/61) for the samples from supermarkets. Of 74 samples from local markets, 28 of them had dimethoate exceeding the MRL value (20 ppb). This represents a rate greater than dimethoate's MRL of 38%. Twenty-two samples purchased from supermarkets were found to contain dimethoate that exceeded the MRL, denoting to a rate greater than dimethoate's MRL of 36%. For diazinon, the occurrence rates in the water morning glory samples were 53% (39/74) and 79% (48/61) for the samples from local markets and supermarkets, respectively (Fig. 3). Only one sample of the water morning glory purchased from the local markets had diazinon exceeding the recommended MRL (10 ppb). None of the water morning glory samples from the supermarkets had diazinon levels above the MRL value. Carbofuran and cypermethrin were detected in the water morning glory samples from both the local markets and supermarkets with moderate occurrence rates. Carbofuran was found in the morning glory samples with occurrence rates of 32% (24/74) and 28% (17/61) for the local markets and supermarkets, respectively. All of the water morning glory samples tested had carbofuran levels below its recommended MRL. Cypermethrin was found in 11% (8/74) of the water morning glory samples bought from the local market samples, while 26% (16/61) of the supermarket samples contained cypermethrin (Fig. 3). Five of the water morning glory samples (7%) from the local markets had cypermethrin that exceeded its MRL (700 ppb). Out of the supermarket samples, four samples (7%) had cypermethrin exceeding the recommended MRL. Chlorpyrifos was also detected in the water morning glory samples from the local markets and supermarkets with low rates of occurrence. It was found in 7% of the water morning glory samples bought from the local market samples, while 21% of the supermarket samples contained chlorpyrifos (Fig. 3). One sample of the water morning glory from both the local markets and the supermarkets had chlorpyrifos that exceeded the MRL (50 ppb). Other six pesticides including carbaryl, chlorothalonil, λ-cyhalothrin, malathion, methomyl and profenofos were detected in the water morning glory samples, although their occurrence rates were very low (Fig. 3). Of note, some of pesticides mentioned were not detected in both the local market samples and the supermarket samples. For example, carbaryl and chlorothalonil were detected only in the supermarket samples, but not found in the water morning glory samples bought from the local markets.

The overall incidence of pesticide detection in the water morning glory samples purchased from local markets and supermarkets. Small proportions of samples were found to contain no pesticide residues; this represents a rate of free of pesticide-free residue of 1% and 3% in the local and supermarkets, respectively. Extremely high percentages of pesticide detection, i.e., 99% and 97% were observed in the morning glory samples bought from the local markets and supermarkets, respectively. The incidence of pesticide residues detected exceeding the recommended MRL values in the morning glory samples from local markets was 42% whereas the incidence rate of 49% was observed in the supermarket samples.

The profiles of pesticides detected in the three vegetables investigated are shown for comparison (Table 1). Details of the pesticides detected in the three studied vegetables and their concentrations are contained in the Supplemental Data. From 28 pesticides studied,

**Table 1  Profile showing types of pesticide residues found in three commonly consumed vegetables studied, Chinese kale, pakchoi and morning glory.**

| Vegetable | Carbaryl | Carbofuran | Chlorothalonil | Chlorpyrifos | λ-Cyhalothrin | Cypermethrin | Deltamethrin | Diazinon | Dimethoate | Malathion | Metalaxyl | Methomyl | Profenofos |
|---|---|---|---|---|---|---|---|---|---|---|---|---|---|
| Chinese kale | √ | √ | √ | √ | √ | √ | √ | √ | √ | √ | √ | – | √ |
| MRL[a] (ppb) | 50 | 20 | 4,000 | 50 | 1,000 | 1,000 | 500 | 50 | 20 | 3,000 | 2,000 | – | 10 |
| Pakchoi | √ | √ | – | √ | √ | √ | – | √ | √ | – | √ | – | √ |
| MRL[a] (ppb) | 50 | 20 | – | 1,000 | 1,000 | 1,000 | – | 50 | 20 | – | 50 | – | 50 |
| Morning glory | √ | √ | √ | √ | √ | √ | – | √ | √ | √ | √ | √ | √ |
| MRL[a] | 10 | 10 | 10 | 50 | 20 | 700 | – | 10 | 20 | 20 | 2,000 | 20 | 10 |

**Notes.**

[a]MRL values for each pesticide in the vegetable studied were cited from recommended MRL values established by *Thailand Ministry of Agriculture Cooperation (2013)*, *Codex Alimentarius Commission (2015)* and *European Commission (2015).*

13 were found in the fresh samples of these three popularly consumed vegetables. Nine pesticides were found to be common pesticides detected in all the three vegetables studied. These were carbaryl, carbofuran, chlorpyrifos, λ-cyhalothrin, cypermethrin, diazinon, dimethoate, metalaxyl and profenofos. Methomyl was not detected in the Chinese kale and pakchoi samples. Chlorothalonil, deltamethrin, metathion and methomyl were not found in the pakchoi samples, while deltamethrin was also not detected in the morning glory samples.

Table 2 shows comparison of pesticide concentrations in the three vegetables studied found in the samples bought from the local markets and the supermarkets. Median data were evaluated and are presented in Table 2. All the pesticide concentrations detected in these vegetables were found to be not normally distributed; therefore, the data was then statistically evaluated by the non-parametric Mann–Whitney test. Subsequently, the median data was used to compare the differences in concentrations of pesticides between the two groups, the local market and the supermarket samples. For the Chinese kale, the median concentrations of dimethoate and profenofos were similar ($P > 0.1$) between the samples from the local markets and the supermarkets. However, the median concentrations of diazinon and metalaxyl in the Chinese kale samples purchased from the supermarkets were significantly greater than those detected in the samples purchased from the local markets ($P < 0.001$, Table 2). With regard to results in pakchoi, the median concentrations of three pesticides, dimethoate, diazinon and metalaxyl in the local market samples were not significantly different from those found in the supermarket samples ($P > 0.05$). Though, the median concentrations of carbofuran in the pakchoi samples bought from the supermarkets were significantly higher than those observed in the local markets ($P < 0.001$, Table 2). For the morning glory samples, there were no significant differences between the samples from the local markets and the supermarkets in median concentrations of pesticides. The exception to this was for the median concentration of diazinon in the supermarket samples was significantly higher than that seen in the local markets ($P < 0.01$, Table 2).

**Table 2  Median concentrations of six commonly detected pesticides in three vegetables studied.**

| Pesticide | Vegetables | | | | | |
| --- | --- | --- | --- | --- | --- | --- |
| | Chinese kale | | Pakchoi | | Morning Glory | |
| | Local markets | Supermarkets | Local markets | Supermarkets | Local markets | Supermarkets |
| Carbofuran | – | – | 0.28 (Med)[a] (0.11, 0.96)[b] (n = 21) | 1.6 (Med)[c] (1.1, 3.5) P = 0.0002 (n = 71) | 8.2 (Med) (2.4, 83.2) (n = 17) | 7.7 (Med) (3.2, 33.4) P = 0.442 (n = 57) |
| Dimethoate | 10.5 (Med) (2.3, 30.1) (n = 57) | 6.4 (Med) (1.9, 15.4) P = 0.17 (n = 50) | 27.2 (Med) (9.8, 46.2) (n = 59) | 18.6 (Med) (4.7, 40.2) P = 0.167 (n = 55) | 10.4 (Med) (3.4, 34.6) (n = 68) | 12.1 (Med) (4.0, 28.8) P = 0.737 (n = 51) |
| Diazinon | 0.28 (Med) (0.13, 1.7) (n = 42) | 2.0 (Med)[c] (0.82, 4.2) P = 0.0004 (n = 41) | 1.4 (Med) (0.67, 2.6) (n = 36) | 1.8 (Med) (1.1, 3.6) P = 0.07 (n = 40) | 1.1 (Med) (0.62, 1.6) (n = 39) | 1.7 (Med)[c] (0.97, 2.7) P = 0.009 (n = 48) |
| Metalaxyl | 0.93 (Med) (0.32, 3.5) (n = 61) | 10.8 (Med)[c] (2.1, 23.9) P = 0.0001 (n = 52) | 8.8 (Med) (2.1, 36.1) (n = 61) | 6.4 (Med) (2.5, 31.4) P = 0.8 (n = 61) | 29.4 (Med) (12.2, 71.3) (n = 27) | 24.8 (Med) (5.8, 64.0) P = 0.562 (n = 27) |
| Profenofos | 9.3 (Med) (0.26, 71.2) (n = 28) | 23.9 (Med) (1.3, 36.3) P = 0.39 (n = 18) | – | – | – | – |

**Notes.**
[a]Median value.
[b]25 and 75 percentiles of the median value.
[c]Statistically significant differences in pesticide concentrations were observed between the local market and the supermarket groups ($P < 0.05$).
 $n$ is number of samples in which the pesticide was detected.

## DISCUSSION

The GC-MS/MS methods established in our laboratory (*Wanwimolruk et al., 2015b*) involving QuEChERS sample preparation and GC-MS/MS analysis were validated. The methods were proven to be suitable and appropriate for determination of pesticide residues in the three leaf vegetables namely Chinese kale, pakchoi and morning glory. This was verified by results of assay validation which have illustrated good recovery, sensitivity, selectivity, linear calibration curves, good reproducibility and accuracy. The utilizations of GC combined with triple quadrupole MS technique not only aided the detection and quantitation of pesticides but it also offered excellent sensitivity for pesticide detection.

The present study examined potential contamination of 28 pesticides in three leaf vegetables namely Chinese kale, pakchoi and morning glory sold in Thailand. Twelve pesticides were detected in fresh Chinese kale samples bought from the local markets and the supermarkets. These included carbamates (carbaryl, carbofuran), organochlorines (chlorothalonil), organophosphorus pesticides (chlorpyrifos, diazinon, dimethoate,

malathion, profenofos), pyrethroids (λ-cyhalothrin, cypermethrin, deltamethrin), and metalaxyl. Findings of so many pesticides detected in this vegetable indicate that pesticides are widely and extensively used in the agronomy of Chinese kale in Thailand. This observation is in agreement with our recent finding in which many pesticide residues were detected in Chinese kale sold in Thailand (*Wanwimolruk et al., 2015b*). Also, this is consistent with those previously observed pesticide contamination in vegetables in Thailand and other Asian countries (*Chang, Chen & Fang, 2005*; *Sapbamrer & Hongsibsong, 2014*; *Swarnam & Velmurugan, 2013*). A study carried out in a northern part of Thailand (*Sapbamrer & Hongsibsong, 2014*) reported that vegetables bought from markets contained organophosphorus pesticides greater than the recommended MRLs. These vegetables included garlic, Chinese cabbage, spring onion, Vietnamese coriander and Chinese kale. These findings with respect to Chinese kale agree with our observation in which the levels of three organophosphorus pesticides (chlorpyrifos, dimethoate and profenofos) were greater than their corresponding MRL values. In the present study, there were five pesticides, namely carbofuran, chlorpyrifos, cypermethrin, dimethoate and profenofos, which were detected in some samples at levels exceeded the MRLs (Fig. 1). This implies that the Thai farmers used these pesticides in excessive doses or did not follow the GAP in which an appropriate pre-harvest interval, i.e., the time period between the last pesticide application and a safe harvest of the treated crop, was observed. Although metalaxyl and diazinon were among the most often detected in the Chinese kale samples, the pesticide residue concentrations found did not exceed their corresponding MRL values. Moreover, finding that a large proportion of the Chinese kale samples (90%) contained multiple pesticide residues (Fig. 1) clearly indicates that Thai farmers are likely to use more than one pesticide during the cultivation of Chinese kale.

Observations on similar types of pesticides detected in these commonly three individual vegetables (Table 1) indicates that Thai farmers cultivated these vegetables in the same areas of their farm, as it is easy to water and protect these vegetables from pests by using the same mixture of pesticides. Another reason could be that Thai farmers producing these vegetables are neighbors and grouped their farms together. Among the pesticides used in cultivation of these three leafy vegetables, metalaxyl, dimethoate and diazinon were the most often used pesticides. Moreover, the similarity in the profiles of pesticides detected in these three commonly consumed vegetables studied suggests that it is an advantage to reduce the cost for the pesticide monitoring by selecting to monitor the pesticide residues in only one of these vegetables. The results from any of these three vegetables will be eventually applied to those of the other two counterparts. Both the extent and incidence of pesticide contamination observed in each vegetable were similar between the samples from both the local markets and the supermarkets. For instance, most of the twelve pesticides found in the Chinese kale were detected in samples from the local markets and the supermarkets. The exception was three pesticides detected in the Chinese kale samples were not found in the samples from both sources. Chlorothalonil and deltamethrin were not detected in the local market samples, while a few samples from the supermarkets were contaminated with residues of these pesticides. Malathion was found in only one Chinese kale sample from the local markets but not in the samples from the supermarkets. Similar findings were

seen in the other two vegetables (pakchoi and morning glory) regarding minor differences in pesticides detected in the samples from the local markets and the supermarkets. These minor differences in the profiles of pesticides found in the three commonly consumed vegetable samples from the local markets and the supermarkets may be related to the sources (or farms) where the vegetables were cultivated, difference of usage of each type of pesticides, and ignorance of GAP awareness. Traceability of the produces was hard to attain and ultimately this was not the primary goal of the current study. If available, the traceability would have been very useful for interpretation of the data obtained from this study. The merchants were asked where they bought the vegetables from and habitually many of them did not have an answer. For those who provided an answer, it appeared that most of the vegetable samples tested were bought from four different wholesale markets in Bangkok and the Nakhon Pathom province near Bangkok. Future studies are required to trace the farms where the vegetables are cultivated and to identify the factors or farmers' behaviors that are attributable to the differences in rates of pesticide detection and MLR exceedance. Vitally, proper education such as GAP regarding the appropriate use of pesticides must be provided to these farmers.

The present study revealed overall incidence of pesticide detection in the three vegetables studied was in a range from 97–100%. For the Chinese kale, this high incidence of pesticide detection is consistent with our previous study published recently (*Wanwimolruk et al., 2015b*). In that study, an incidence of pesticide detection of 85% was reported in Chinese kale collected in Nakhon Pathom province of Thailand. Characteristics and sources of the samples were similar to those tested in the present study. It is obvious that these figures of the incidence of pesticide detection observed in the three commonly consumed vegetables are noticeably higher than the tolerable detection rate in western or developed countries, such as USA and European Commission (EC) countries like France, U.K., Norway and Germany. For example, the US FDA carried out a monitoring program of vegetables with thousands of domestic samples and imported samples (*Granby et al., 2008*). Pesticide residues were found in 30% of the domestic vegetables and 21% of the imported vegetables. In Taiwan, pesticide residues were detected in 14% of 9,955 vegetable samples tested (*Chang, Chen & Fang, 2005*). A survey study conducted in India found residues of many organophosphorus pesticides (e.g., chlorpyrifos, dimethoate, monocrotophos and profenofos) in 54% of the vegetable samples (*Swarnam & Velmurugan, 2013*). The latest study from Thailand (*Sapbamrer & Hongsibsong, 2014*) conveyed an overall pesticide detection rate of 25% ($N = 106$) in various vegetables bought from the markets. This rate is nevertheless much lower than the rates of pesticide detection in the Chinese kale, pakchoi and morning glory observed in this study. The difference may be accounted for by differences in seasons of vegetable cultivation, vegetable types, types of pesticides used and analytical methods employed.

Remarkably, the occurrence of pesticide detection exceeding the MRL in the three vegetables studied ranged from 35 to 71%; it was high in both samples from the local markets and the supermarkets. These were noticeably high compared with the incidence testified in developed countries. For instance, the US FDA declared that violations (with pesticide concentration >MRL) were found in 2% of the domestic and 7% of the imported

vegetable samples (*Granby et al., 2008*). The European Union (EU) Monitoring Program for pesticides declared that 5% of vegetable samples examined had pesticide residue concentrations that exceeded the MRL (*Granby et al., 2008*). In Asia, a study carried out in Taiwan reported that of 9,955 samples tested, 1.2% violated the MRL (*Chang, Chen & Fang, 2005*). Therefore, the incidence of pesticide detection of >MRL in our three vegetables, at rates of 32 to 49% are unusually high when compared with acceptable rates reported in developed countries. Nevertheless, these incidence rates are somehow similar to that found in Pakistan, an Asian country, in which 206 different vegetables were analyzed for 24 pesticides, and 46% had levels greater than the MRL (*Parveen, Khuhro & Rafiq, 2005*). Also, a study from India (*Swarnam & Velmurugan, 2013*) reported that 15% of vegetable samples tested contained pesticide residues that exceeded the MLR values. In addition, the incidence of pesticide detection of >MRL was stated to be 24% in several market vegetables examined in northern districts of Thailand (*Sapbamrer & Hongsibsong, 2014*). This rate of pesticide detection is quite comparable to the rates reported in the present study. Recently, the Food and Drug Administration (FDA), Ministry of Public Health of Thailand issued a report on the pesticide monitoring program for vegetables and fruits in which more than 60,000 samples were screened each year (*Srithongkum, 2014*). The report revealed that violations found in vegetables and fruits marketed in Thailand were in a range of 5% in the year 2011 to 4% in the year 2013. These rates reported by the Thailand FDA were approximately 7–14 times lower than the incidence of pesticide detection exceeding the MRL (35–71%) found in this study. The conflicting findings are likely to be accounted for by the difference in methods utilized in the two survey studies. The survey by the FDA of Thailand was done by using a cholinesterase inhibition assay kit called GT-Test kit. This assay kit is capable of detecting two groups of pesticides, i.e., carbamates (carbofuran and methomyl) and organophosphates (dicrotophos and EPN). Nevertheless, unlike our current GC-MS/MS method, the GT-Test kit cannot offer a quantitative analysis like most analytical methods, such as UV spectrophotometric assays, LC-MS/MS and GC-MS/MS. Because the kit assay is restricted in detection to only four individual pesticides, it has less sensitivity and does not provide a quantitative determination of pesticide concentration. Thus, these restrictions of the kit assay can underestimate the incidence of MRL violations.

Unusually high rates of exceedance of the MRL found in the three vegetables investigated may be due to the fact that we used the recommended MRLs adopted from those employed in developed countries, i.e., *Codex Alimentarius Commission (2015)*, and *European Commission (2015)*. Some of MRLs for pesticides used may be too low and made the incidence unnecessarily high. For examples, MRL values for carbofuran were 20 ppb (0.02 ppm) for Chinese kale and pakchoi; and 10 ppb (as a default value) for the morning glory. The MRLs for profenofos were 10 ppb (0.01 ppm) as a default value, for Chinese kale and morning glory; and 50 ppb for the pakchoi. Using these low recommended MRLs yielded the remarkably high rate of MRL exceedance observed in the present study. In addition, it should be noted that our findings were limited to these three vegetables sold in a small number of central provinces of Thailand and did not reflect the figure for the whole country. This is because the sample sizes were considerably rather small. Larger sample sizes collected from many provinces of different regions in Thailand would be required to

verify the incidence of pesticide contamination. Importantly, health risk assessment due to consuming these pesticide contaminated vegetables has not yet been evaluated. A larger sample size would be necessary for that as well.

There were substantial variations in the levels of pesticides found in the three vegetables studied. For instance, profenofos levels found in the Chinese kale samples varied widely among the samples from both sources ranging from 0.1 to 2,095 ppb; and levels of carbaryl found in these samples ranged from 0.1 to 606 ppb. In addition, the large S.D. values (relative to their corresponding means) were found for each pesticides detected in the Chinese kale and also in the other two vegetables. This reflects the huge variation in concentrations of each pesticide detected in the three commonly consumed vegetables. The large variation in the level of pesticides detected in the vegetables may be due to many factors influencing the pesticide residues that remained on the vegetables at the time of harvest. These factors include the dosage of pesticides applied, dosing frequency and the pre-harvest interval for crops (*Banerjee et al., 2006*; *Zhang et al., 2012*). Appropriate education on pesticide use and the pre-harvest interval for crops is necessary. This education will assist to reduce the amount of pesticide residues remaining in vegetables and fruits.

Critically, the remarkably high rate of exceedance of the MRL (ranged from 35 to 71%) found in the three commonly consumed vegetables reported in the present study indicates that these vegetables either purchased from both the local markets and the supermarkets are highly contaminated with pesticide residues. Regarding Thai people's expectations of supermarket produce, the findings in this study raises the question as to the quality of the vegetables sold in supermarkets in Thailand. The quality of vegetables sold in the supermarkets in Thailand is, in general, thought to be good with regard to levels of pesticide contamination. Thai people's perception of supermarket vegetables and fruits is high with respect to quality and freshness. Most Thai consumers believe the labels placed on the produce sold in the supermarkets in which they are claimed to be pesticide-free or organic produce. However, these labels and claims are made without scientific evidence and testing to support them. The quality, in terms of pesticide contamination of vegetables sold in the local markets in Thailand is not guaranteed, as the routine national monitoring programs of pesticide residues is not fully implemented (*Wanwimolruk et al., 2015b*). The existing evidence points to considerable food safety problems, since pesticide residues were noticeably detected in vegetables sampled from the local markets in Thailand (*Sapbamrer & Hongsibsong, 2014*; *Wanwimolruk et al., 2015b*). Such quality of these three commonly consumed vegetables sold in Thailand appears to be similar regardless of where the vegetables are purchased from, i.e., from local open-air markets or supermarkets. The present study has also demonstrated that there was similarity in the profiles of pesticides detected in the three commonly consumed vegetables from these two sources. In addition, the current study did not aim to compare organically grown and conventional grown vegetables but rather to compare the quality of the three commonly consumed vegetables bought from local markets and from supermarkets, in term of pesticide contamination. As previously mentioned, the vegetables sold from supermarkets of Thailand were categorized into four groups, i.e., conventional, organic, pesticide-free and safe vegetables. However,

most vegetables sold in the supermarkets have claimed to be either organic and pesticide-free produce. Such statements issued by the supermarkets are not always reliable. Our study did not test all organically grown vegetables in Thailand so the findings are limited to the three vegetables studied. Future studies are warranted to verify if the produce sold in the supermarkets claiming to be organic and pesticide-free are of better quality than the conventionally grown produce (in terms of pesticide contamination).

The prices of vegetables and fruits sold in supermarkets in Thailand are substantially higher (2–6 times) than the produce sold in the local open-air markets. For example, the average price of Chinese kale from supermarkets was 112 ± 44 Bahts/kg, (approximately US\$3.4/kg) which was more expensive than those from local markets (38 ± 8 Bahts/kg, US\$1.1/kg). In spite of this, for some pesticides such as diazinon and metalaxyl, the levels of these pesticides in the Chinese kale samples from the supermarkets were significantly higher than those seen in the samples from the local markets (Table 1). A similar observation was also found in the other two vegetables investigated, pakchoi and morning glory. This implies that the level of pesticide contamination of these three commonly consumed vegetables cannot be warranted by the price of the produce. However, it may be correct that vegetables and fruits purchased from the supermarkets are fresher than those from local open-air markets.

Our findings also emphasize the fact that these three commonly consumed vegetables, namely Chinese kale, pakchoi and morning glory, sold in the supermarkets in Thailand are not pesticide-free or organically grown as the merchants state on the produce labels. This problem is challenging the Thai government authorities such as the Thai FDA and the Department of Agriculture. The financial sponsor of this study, the Agricultural Research Development Agency (Public Organization) of Thailand requested that we, as researchers, to disseminate our findings through the Thai government authorities in order to facilitate the implementation of regulations and laws on pesticide residues and food safety. The Thai government authorities have been informed about the findings raised from this study. Further action has been planned to rectify the situation with the supermarket stakeholders, continue pesticide monitoring program, reinforce the laws, and properly instate the GAP system to the farmers. These are very important not only to reduce the health risks of consumers associated with pesticide residues in vegetables but also to protect consumers' rights. The consumers who buy produce labeled organic pay more so should get a higher quality, pesticide-free produce.

## CONCLUSIONS

There is considerable contamination of pesticides in the three commonly consumed vegetables in Thailand, i.e., Chinese kale, pakchoi and morning glory. Nine to twelve pesticides were detected in these vegetables at detection rates of 97–100%. The rate of pesticide residues exceeding the MRL in these vegetables studied were remarkably high as compared to those reported in developed countries. The incidence of pesticide contamination was found to be similar between the vegetables bought from local markets and supermarkets. These findings questioned the quality of vegetables claimed to be

pesticide-free sold in the supermarkets and urged the Thai government authorities to solve this important problem. This conclusion excludes Thai export quality vegetables and fruits that are routinely monitored for pesticide contamination before exporting. It is our recommendation to the Thai government authorities to conduct a proper pesticide monitoring program for these three commonly consumed local vegetables to protect the health of domestic consumers. The findings from this study would also be useful for the Thai government to ascertain the MRL of pesticides in these three commonly consumed vegetables, and to incorporate other pest management strategies for the safe and appropriate use of pesticides.

## ACKNOWLEDGEMENTS

We are thankful to Dr. Nipa Rojroongwasinkul, Institute of Nutrition, Mahidol University for her guidance in the statistical analysis, and Ms. Onnicha Kanchanamayoon for her help in GC-MS analysis.

### Funding

The authors received financial support from the Agricultural Research Development Agency (Public Organization), Thailand, and International Relations for Research Section, Division of International Affairs, National Research Council of Thailand. The funders had no role in study design, data collection and analysis, decision to publish, or preparation of the manuscript.

### Grant Disclosures

The following grant information was disclosed by the authors:
Agricultural Research Development Agency.
International Relations for Research Section, Division of International Affairs, National Research Council.

### Competing Interests

The authors declare there are no competing interests.

### Author Contributions

- Sompon Wanwimolruk conceived and designed the experiments, performed the experiments, analyzed the data, contributed reagents/materials/analysis tools, wrote the paper, prepared figures and/or tables, reviewed drafts of the paper.
- Kamonrat Phopin and Somchai Boonpangrak prepared figures and/or tables, reviewed drafts of the paper, collected samples.
- Virapong Prachayasittikul reviewed drafts of the paper.

### Data Availability

  The raw data has been supplied as Supplemental Files.
## Supplemental Information

Supplemental information for this article can be found online at http://dx.doi.org/10.7717/peerj.2432#supplemental-information.

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
