# Peer review of "Food safety in Thailand 4: comparison of pesticide residues found in three commonly consumed vegetables purchased from local markets and supermarkets in Thailand"

_PeerJ, doi:10.7717/peerj.2432_

## Round 0.1 · original submission · Minor Revisions

· Academic Editor

Minor Revisions

Dear authors,

This is an important contribution that demonstrates with high confidence the presence of pesticides in the most preferred, supposedly organic foods, by people in different countries. Possibly this issue occurs most frequently in developing countries, although I am not sure that the evidence that you provide is enough to confirm that pesticide rates estimated for organic food are lower than in developing ones.

The manuscript is well constructed and the methodology and conclusions seem to be correct, although a brief description of the data treatment should be included. The manuscript can be substantially improved following the recommendations from the reviewers and those I suggested in the attached, annotated pdf. Please, consider them and modify the text accordingly. As you will see, the main concerns are related to the redundancy in the discussion section, which you can fix avoiding repetition of data that were extensively explained in the Results section. Moreover, the inclusion of a table showing the main results in a way that they can be easily depicted, is convenient. Doing so, you can reduce the number of figures that then, will be unnecessary. You also have to check the data in Table 2.

·

Basic reporting

1. Please clarify what the MRL (in the abstract) is.
2. The reference should be added for MRL when the authors mentioned ">MRL".
3. Citing Wikipedia does not sound scientific. Please other reference, in this case the original work should be cited (line 98).
4. Line 324, please check if it is morning glory or pakchoi sample.

Experimental design

No comment.

Validity of the findings

1. Figures 2, 4, and 6 should be omitted. The data can be found in Figures 1, 3 and 5, respectively. The authors may summarize in the Table form, if they feel that the data should be presented. Moreover, the % sample of not detected, when sum with <MRL and >MRL, is not the same as total detected. Please also check this point.
2. In Table 1, the MRL may be added so that the audience can understand easily.
3. In Table 2, please check the data again. It seems that something is wrong. Why the S.D. is very high in some cases? Did the authors mistype?
4. In Table 2, why both the mean and median are shown? The authors discussed only one of them. And why should the median at 25% and 75% be presented?

Additional comments

The results are interesting. However, there are some points are needed to be addressed or corrected.

Reviewer 2 ·

Basic reporting

Despite the article seems to have a lot of work it is very difficult to evaluate it without having any details of the developed and validated methodology that seems it was published in another journal. Almost the on going validation data must be included in the manuscript. The validation data should be clearly reported for each analyzed commodity.

Experimental design

The data treatment should be more exhaustive. The frecuencies of the findings are not enough to have a strong conclusion in order to be published. The pesticides concentrations found in the performed analysis must be reported.

Any tables of results should be presented for better comprehension of them.

What vegetables are analyzed and the specifics results for each one should be also reported or described in a table, Each commodity should be correlated with each concentration found and compared with their MRLs. The discussion for the MRLs should be clarified and clearly discussed between MRLs fixed by Codex and by EU.

Validity of the findings

The findings are valuable as monitoring program of a country for different vegetables highly consummed.

To give more value to the work the suggestion gave above should be included in the manuscript.

Additional comments

The paper is interesting but more experimental data should be given and the results should be more clearly discussed in order to fully understand and highlight the work already done.

Reviewer 3 ·

Basic reporting

No comments

Experimental design

Line 68 = via food, for toxicological term is oral ingestion

Line 77 nor and either, I could not find what was being compared

Line 78 in spite of herbicides which are the most commercialized, only one was evaluated and It was not found, is important to remark

Line 89 This possesses potential toxicity if vegetable is eaten freshly and daily, this?? it is not clear that the presence of pesticides in fresh vegetables is an important health risk concern

Validity of the findings

L116 “organic vegetables” it is not clear, all products collected in supermarkets are organic; in such way all the samples from local market are conventional?
The work compared organic (from supermarket) and conventional (from local market)?
The supermarket sells only organic vegetables?
The suppliers from supermarket are different from local market?
The farmers from local market do not sell their products to the supermarket?

In such way, as the results are quite the same it is possible to conclude that farmers distribute their products to supermarket and also to the local market

Additional comments

The results and discussion are very good, but I think that is too extensive, if possible simplify or reduce some subjects that have being sited before.

---

## Round 0.2 · Minor Revisions

· Academic Editor

Minor Revisions

Dear authors,

The information contained in this manuscript deserves to be known. However, and although the manuscript has been improved by addition of the useful suggestions made by the reviewers, some concerns remain.

1-The main problem in my concept as your Academic Editor for PeerJ is the confusing, and apparently not real separation that you pretend to remark between the samples from the local markets vs. those from the supermarkets. You described such separation at the beginning of the manuscript explaining that you purchased samples of specific vegetables that you want to analyse for detection of pesticides following this concept: conventional from the local markets and organic and/?or free of pesticides from the supermarkets. Thus, the initial association of the readers (and one of the reviewers) is that supermarkets only sell conventional produces, which we realize that it is not true when continuing reading the manuscript. To my and the reviewer 3 concern, it must be necessary that you clarify this issue in the introduction, leaving clear that you also purchased and analysed organic samples from the supermarkets, as well as conventional samples bought from the local markets (see also the annotated manuscript that I am uploading).

2-Even though the organic production in developed countries appears to be strictly regulated and the producers must follow several specific standars (e.g. USDA regulation for organic certification in USA), chemicals can be found on food labelled as “organic”. This is possible because the crops can be sprayed with chemicals not exceeding an established proportion of the product weight. Thus, despite labelled as organic you can find pesticides in these products, although not in an almost equivalent concentration than they are found in conventionally produced food. Organic crops can also be contaminated by drifting of pesticides from neighbouring conventional farms. But If that happen, the concentration of pesticides found would be comparatively lower. It may be interesting that you include some comment about this, considering that you noted different criteria to establish the MRL in developing and developed countries.

You can find more information from:
Dangour AD, Dodhia SK, Hayter A, Allen E, Lock K, Uauy R. Nutritional quality of organic foods: a systematic review. American Journal of Clinical Nutrition 2009; 90:680-5.

EFSA 2009. Pesticides used in organic farming: some pass and some fail safety authorization. European Food Safety Authority (EFSA). Available from: www.ecpa.eu (Viewed 19 Nov, 2009).

Bier AH. (2008). Organic Standards for Crop Production. National Center for Appropriate Technology (NACT). National Sustainable Agriculture Information Service (ATTRA), U.S. Department of Agriculture (USDA), Washington, D.C.

SCIENTIFIC REPORT OF EFSA. 2013. European Union report on pesticide residues in food European Food Safety Authority. European Food Safety Authority (EFSA), Parma, Italy. EFSA Journal 2015;13(3):4038.

3-The discussion remains reiterative; you can fix it by deleting some sentences. I have highlighted some in the annotated pdf, where you will find some other comments and suggestions that I would like you pay attention. One of them is the organization of the tables. In my opinion, tables that were provided as Supplementary Information should be part of the main text, maybe replacing Table 1. You solely need to add a new column to the right where the MRL for each analysed pesticide should be provided. In that way, it might be easier for the readers to compare the detected proportions of pesticides with these established as minimal allowed by the different regulatory agencies.

4-Please, consider to make these arrangements and resubmit the manuscript for a new revision, or provide a reason for an eventual rebuttal.

Best regards,
Graciela Piñeiro

---

## Round 0.3 · Minor Revisions

· Academic Editor

Minor Revisions

Dear authors:

I am very glad to see that you accepted my suggestions and consider them as useful. The manuscript is almost ready to be accepted. I just need that you fix the text as I have tracked in the manuscript attached. Please, pay attention to the references, one of them was not included in the list. I look forward to receive the revised version soon.

Best regards,
Graciela Piñeiro

---

## Round 0.4 · accepted · Accept

· Academic Editor

Accept

Dear Dr. Wanwimolruk and collaborators,
It is my pleasure to let you know that your manuscript “Food safety in Thailand 4: Comparison of pesticide residues found in three commonly consumed vegetables purchased from local markets and supermarkets in Thailand” is now ready for publication in PeerJ. This is a very important contribution and I am very glad to have had the opportunity to handle it. Congratulations!
Yours sincerely,
Graciela Piñeiro